# Exploratory Meta-Analysis of the Effect of Malic Acid or Malate Addition on Ruminal Parameters, Nutrient Digestibility, and Blood Characteristics of Cattle

**DOI:** 10.3390/ani15152177

**Published:** 2025-07-24

**Authors:** Leonardo Tombesi da Rocha, Tiago Antonio Del Valle, Fernando Reimann Skonieski, Stela Naetzold Pereira, Paola Selau de Oliveira, Francine Basso Facco, Julio Viégas

**Affiliations:** 1Animal Science Department, Federal University of Santa Maria, Avenida Roraima 1000, Santa Maria CEP 97105-900, Brazil; tiago.valle@ufsm.br (T.A.D.V.); stela.pereira@iffarroupilha.edu.br (S.N.P.); paolaoliveiraselau@gmail.com (P.S.d.O.); francine.facco@acad.ufsm.br (F.B.F.); julio.viegas@ufsm.br (J.V.); 2Animal Science Department, Federal University of Technology-Parana, Estrada para Boa Esperança km 04, Dois Vizinhos CEP 85660-000, Brazil; fskonieski@yahoo.com.br

**Keywords:** blood metabolites, dietary covariates, meta-regression, organic acid, ruminal pH, volatile fatty acids

## Abstract

Improving nutrient use and metabolic balance in cattle is essential for sustainable livestock production. Malic acid, a natural compound, has been studied as an alternative to conventional feed additives. This study combined data from several experiments to evaluate the effects of malic acid and malate on digestion and blood indicators in cattle. Malate improved ruminal pH stability, supporting fiber digestion. Both compounds increased beneficial fermentation products, especially propionate, and improved blood glucose while reducing signs of fat mobilization. These findings suggest that malic acid and malate can enhance digestive efficiency and energy balance in cattle, offering a natural strategy to improve performance and reduce reliance on synthetic additives.

## 1. Introduction

Feed additives have been used to improve ruminant productivity for decades [1]. However, one of the most popular feed additives (ionophores) has faced restrictions due to the potential limited safety [2]. In this way, several additives are evaluated as ionophore replacers. Among these alternatives are essential oils, tannins, enzymes, probiotics, and organic acids. These compounds aim to modulate rumen fermentation, improve nutrient digestibility, and enhance animal performance.

Within this context, malic acid (MAC) emerges as a promising additive. It is a citric acid cycle intermediate, and its addition increases in vitro rumen lactate uptake, volatile fatty acids (VFA) production, and diet digestibility, reducing methane emission [3,4,5]. In addition, MAC increases in vivo VFA rumen concentration [6,7] through increased propionate and butyrate content [8]. In addition, animals fed MAC show higher rumen pH [9] and improved nutrient digestibility [10]. However, these effects have not been observed in all studies [7,11,12].

The diversity of results among studies may be linked to differences in MAC presentation (acid or salt) and potential interaction with the substrates, such as dietary starch level [5]. In this sense, meta-analysis could be used to combine the results of several experiments into a single effect estimate to determine the real effect of the MAC on the variables of interest, in addition to determining and quantifying the influence of covariates on the meta-analyzed result.

We hypothesized that true effects associated with MAC addition, particularly on rumen pH, lactate uptake, and propionate production, could be identified and better explored through meta-analysis. Additionally, we expected that qualitative and quantitative covariates related to diet composition and MAC presentation would be useful to explain the between-study variation. This approach would allow the identification of response variables potentially affected by MAC and the covariates that may influence or direct these effects, an aspect that is not well demonstrated in the current literature.

Therefore, the objective of this study was to determine the overall effects of MAC addition on ruminal parameters, nutrient digestibility, and blood metabolites in cattle, and to identify and quantify potential sources of variation in these responses using exploratory analyses.

## 2. Materials and Methods

### 2.1. Database

The manuscript search was carried out in the search engines “Web of Science”, “Science Direct”, and “Google Scholar”. The Boolean moderators used, alone or in combination, were: “organic acids”, “malic acid”, “malate”, and “bovine”. Searches were based on the title and abstract of papers published between 1980 and 2023. The PICO (population/problem, intervention/exposure, comparison, outcome) method was considered to build the database [13]. The population was cattle; intervention was the addition of MAC acid or salt; control was cattle fed without MAC; and results were rumen parameters, nutrient digestibility, and blood parameters.

The studies needed to be original and show the mean and dispersion for each variable. As the analysis requires the standard deviation associated with each variable, if these were not provided directly, they were calculated using the measures presented in the paper, such as standard error of the mean, coefficients of variation, and others. All the transformations followed the recommendations of the Cochrane Handbook for Systematic Reviews of Interventions (Chapter 6.5). Only studies that presented results for a control and treatment group (MAC addition) were considered. Independent experiments in the same study were included as a new comparison. Similarly, malate doses in the same trial were included as a new comparison. Studies in which the inclusion dose of malate was unclear, or where malate was tested alongside other additives without an appropriate control, were not included in the database. The following information was recorded from each trial: study reference, adaptation period, year, experimental design, MAC dose, forage-to-concentrate ratio, diet chemical composition, dry matter intake, animal breed, animal category, and initial body weight. Additionally, the chemical form and commercial source of the malic acid or malate used in each study were also recorded. While not all products were chemically identical, most additives were either DL-malic acid (acid form) or buffered salts such as sodium or calcium malate. Only studies reporting the diet’s chemical composition or providing information to estimate it were included (Table 1). Rumen parameters, nutrient digestibility, and blood parameters results were recorded. The final database had 34 comparisons from 13 studies. Studies involved heifers (n = 2), steers (n = 4), dairy cows (n = 6), and calves (n = 1), covering both beef and dairy animals, with initial body weights ranging from 122 to 895 kg. This variation was not restricted, as the objective of the meta-analysis was not to assess performance-related variables, regardless of animal category. At this point, it is important to note that, due to the limited number of studies, the results obtained in this meta-analysis should be considered exploratory, as the small evidence base may undermine statistical power and the observed heterogeneity may limit broader inferences.

### 2.2. Statistical Analysis

The effect of MAC inclusion was evaluated using the effect size (ES) to measure the treatment effect. The ES was calculated as the difference between the treatment group (MAC acid or MAC Salt) and the control group, divided by the pooled standard deviation of each trial. The average effect of MAC addition was calculated using the “DerSimonian and Laird” random effects model [21]. Heterogeneity across trials was checked using Cochran’s Q test, according to Higgins et al. [22].

Two analyses were performed to discuss the results: subgroup comparison and meta-regression. Subgroup analysis was performed by dividing the studies into two groups using the MAC form: acid vs. salt. Meta-regression was used to explore the linear effects of covariates as variability sources. The following covariates were evaluated: NDF intake (g/kg BW), ADF intake (g/kg BW), starch intake (g/kg BW), and MAC intake (g/kg BW). It is important to point out that only those variables that had at least 10 comparisons and significant heterogeneity were subjected to meta-regression analysis [22]. Forest plots were used to present the average effect size and confidence interval. The leave-one-out sensitivity analysis was applied to assess the influence of individual studies on the overall effect size estimated in the meta-analysis. This method systematically excludes one study at a time and recalculates the pooled effect to determine whether the exclusion of any single study substantially alters the results. This approach is important for identifying influential outliers and ensuring that the meta-analytic conclusions are not driven by a single study. All analyses were performed using the OpenMetaAnalyst statistical software (version 3.13), which operates based on the “metafor” package in R (version 1.9-9).

## 3. Results

### 3.1. Rumen Parameters

MAC addition did not affect cattle rumen pH (overall ES = 0.310, *p* = 0.17). In addition, subgroup analysis showed that MAC acid also had no effect (*p* = 0.12) on pH. The leave-one-out analysis showed that no individual study significantly altered the overall effect on rumen pH (SMD = 0.347, *p* = 0.126), indicating that no influential or interfering studies were present (Figure 1). However, MAC salt increased rumen pH (subgroup ES = 1.420, *p* < 0.01). No effect of addition (*p* > 0.05) was observed for ammonia nitrogen (NH_3_N) (Table 2).

The heterogeneity between studies (I^2^) associated with these variables was significant, being 36.62 and 72.95% for NH_3_N and pH, respectively. Among the covariates tested in the meta-regression, NDF intake decreases (*p* ≤ 0.01) the ES of MAC addition for rumen pH and ammonia-N concentration (Table 3).

In general, MAC salt or acid had no effect (*p* > 0.05) on rumen acetate, butyrate, and lactate (Table 4).

Rumen propionate proportion (overall ES = 0.560, *p* > 0.01) and total VFA concentration (overall ES = 0.508, *p* = 0.03, Figure 2) were increased by MAC addition. Considering the subgroups, MAC acid increased (*p* = 0.02) rumen propionate, whereas MAC salt tended to increase (*p* = 0.08) in the same variable. The leave-one-out analysis for rumen propionate (Figure 3) confirmed that no individual study was interfering or overly influential, as the overall effect remained significant (SMD = 0.610, *p* = 0.003) and stable across all iterations. MAC addition reduced the acetate to propionate ratio (overall ES = −1130, *p* > 0.01). The between-studies heterogeneity was significant and greater than 65% for all variables related to VFAs and lactate in rumen. The meta-regression showed that the main covariates that affect the ES of MAC addition for acetate, propionate (Figure 4), and lactate were starch, starch, and NDF intake, respectively. Acetate to propionate ratio covariates were (*p* ≤ 0.05) NDF and starch intake. None of the covariates were useful in explaining the variation in the MAC effect on butyrate proportion.

### 3.2. Digestibility

In general, MAC increased (*p* ≤ 0.05) macronutrient digestibility with ES of 0.547 for DM, 0.422 for CP, and 0.635 for ADF in total-tract apparent digestibility. However, MAC did not affect (*p* > 0.05) OM and NDF apparent digestibility. Dry matter digestibility increased (*p* ≤ 0.05) in animals fed a MAC acid and was not affected (*p* > 0.05) by MAC salt. However, MAC acid had no effect (*p* > 0.05), and MAC salts increased (*p* ≤ 0.05) CP and NDF digestibility. The MAC acid tended (*p* = 0.08) to increase ADF and the MAC salts increased (*p* = 0.03) ADF digestibility. The leave-one-out analysis for CP digestibility showed a consistent and significant effect across studies (SMD = 0.448; *p* = 0.005). Removal of any individual study resulted in minimal changes to the effect size (ranging from 0.392 to 0.512) and maintained statistical significance (all *p*-values < 0.05). Heterogeneity was significant (*p* < 0.05) for most variables related to digestibility, except for CP and OM. The NDF intake tended to decrease the ES of MAC on DM digestibility. Starch intake was the main covariate explaining the variation in MAC’s effect on NDF digestibility.

### 3.3. Blood Parameters

There was an overall effect of MAC on blood glucose concentration (ES = 0.170, *p* = 0.05). In addition, MAC decreased blood non-esterified fatty acids (NEFA) (overall ES = −0.404, *p* = 0.03). On the other hand, MAC showed no effect (*p* > 0.05) on blood ß-hydroxybutyrate and lactate. However, MAC acid decreased lactate in blood (subgroup ES = −1.661, *p* > 0.01). Despite the significant heterogeneity for most of the blood parameters, the small number of studies was a limiting factor for carrying out the meta-regression analysis.

## 4. Discussion

### 4.1. Rumen Parameters

We hypothesized that true effects associated with malic acid addition, particularly on rumen pH, lactate uptake, and propionate production, could be identified and better explored through meta-analysis. These primary effects would lead to secondary ones on the diet digestibility and blood metabolites. Additionally, we expected that qualitative and quantitative covariates related to diet composition and MAC presentation would be useful to explain the between-study variation. In this context, our hypotheses were at least partially confirmed. Significant effects were observed for some of the main variables where the interference from the MAC addition was expected. Subgroup analysis indicated that the chemical form of the additive may be decisive for the control of rumen acidity. Additionally, the meta-regression indicated that there are dietary covariates that significantly influence the effect size of MAC addition on some outcomes.

For rumen pH, the effects observed for the MAC presentation make sense if we consider the chemical nature of the feed additive. In vitro studies indicate free malic acid and its disodium salt have similar effects, except for the reduction in pH caused by free malic acid [3]. The effect of acidic MAC observed in our study, although small (ES = −0.310) and not significant, indicates that its use negatively impacts rumen pH. On the other hand, the ES of 1.420 observed in the salt subgroup is considered large [23]. Effect sizes greater than 0.8 are considered large according to the Cohen scale. However, it is important to point out that the scale is subjective, and the context in which it is being applied must be considered. Additionally, this ES indicates that the mean pH of the control group and the malate group are separated by 1.42 standard deviations. Considering a standard deviation of 0.18 for pH (average from studies in the database), MAC salt addition would increase pH by 0.26 units. One of the premises that led malic acid to be tested in ruminant diets was its ability, demonstrated in vitro, to increase lactate uptake [4]. The presence of this organic acid favors the growth of *Selenomonas ruminantium*; these bacteria use lactate as a source of carbon and energy, which would imply maintaining rumen pH [5]. Additionally, MAC may act on pH through a second mechanism, which is the production of CO_2_ by *S. ruminantium* [9]. The I^2^ values indicated that 80.4% of variability occurred due to differences between studies. Values of I^2^ higher than 30% represent substantial heterogeneity, which may be investigated [22]. It was observed that NDF intake reduces the mean difference observed between treatments. It is possible that NDF intake ends up shadowing the effect of MAC on ruminal pH, as the presence of NDF implies longer rumination time and, consequently, greater buffering of ruminal pH [24].

Although there was an increase in rumen pH in animals fed MAC, it was not possible to confirm the effects of MAC addition on lactate uptake in the rumen; even though the direction of the effect indicates a likely reduction (overall ES = −0.113), especially when using the free acid (ES = −0.621), the result was not significant. The meta-regression analysis indicated that NDF intake decreases the ES, that is, in studies where NDF intake is higher, the effect of MAC on lactate is smaller. High NDF intake implies low lactate levels in the rumen, impairing the growth of *S. ruminantium* and, consequently, the MAC effect. Additionally, for studies with high fiber intake, depending on the type and form of forage used, MAC addition may be occurring close to or above the limit at which its effect reaches a plateau. Malic acid can represent 2.2 to 4.5% of the dry matter of grasses and 2.9 to 7.5% of legumes, with this amount decreasing with the plant maturity [25]. Furthermore, preserved forages such as hay and silage have a lower content of this component. At this point, it is important to highlight that covariates such as fresh forage, dry forage, and conserved forage intake, as well as the forage:concentrate ratio, were tested as continuous (meta-regression) or categorical (subgroup) covariates but were not significant.

The effect size (0.508) obtained for total VFA concentration is considered moderate according to Cohen’s [23]. This is an expected response and occurred mainly due to the greater production of propionate, as there were no changes in the proportion of acetate and butyrate in the rumen. Concomitantly, the acetate:propionate ratio was higher for the control group, which confirms the higher proportion of propionate (overall ES = 0.560) in the animals receiving MAC. If we consider an average standard deviation of 4.84 for the molar proportion of propionate, this ES may represent a difference of 2.71 percentage points between the means of animals fed diets with or without MAC. The higher propionate production occurs because *S. ruminantium* bacteria can use lactate as a carbon source, provided that oxaloacetate precursors such as malate are present [26]. This acid can follow the reverse cycle of the succinate-propionate pathway and provide the oxaloacetate for lactate fermentation to propionate [5]. The heterogeneity values showed that there is high variability that is not associated with chance. The response indicated by the meta-regression seems in line with what is known about the mechanisms of action of MAC, since starch intake favors the growth of lactate-producing bacteria, which, associated with malic acid, becomes a substrate for the production of VFAs by *S. ruminantium* [27].

The analysis of the NH_3_N indicated that MAC addition, regardless of the form, results in negligible effect sizes (overall ES = 0.079). This variable can be a marker of the amount of N available for synthesis and/or absorption in the rumen [28]. This N, when used to increase the microbial population in the rumen, would culminate in greater bacterial fermentation, which, ultimately, could increase the digestibility of ruminant diet fractions. The meta-regression pointed out the NDF intake as a possible interfering factor in the effect of MAC on NH_3_N. Increased fiber intake stimulates an increase in cellulolytic bacterial populations, while those with proteolytic and amylolytic characteristics decrease [29]. Furthermore, the NDF intake can reduce the concentration of sugars in the rumen [30], which is one of the substrates for *S. ruminantium*, a malate-utilizing bacterium.

### 4.2. Digestibility

The results obtained for fiber and protein digestibility may be a secondary effect of MAC on the control of acidity in the rumen, since the drop in ruminal pH reduces the degradability of fibrous fractions and protein [28,31]. The association of this effect with pH would also explain why the effects on NDF, ADF, and protein digestibility were only detected when the MAC salt (malate) was used, since the acid form was not shown to affect rumen pH. Additionally, MAC can remove H_2_ from the rumen, stimulating an increase in the population of cellulolytic bacteria, which ends up impacting the total digestibility of fibrous fractions [32]. Furthermore, the effect observed on protein digestibility may have occurred due to the increase in the activity of proteolytic enzymes and/or a decrease in the duodenal pH necessary for effective proteolytic activity, promoted by malic acid [33,34]. The increase in DM digestibility due to MAC inclusion may occur due to an increase in enzymatic activity, increased secretions, and association with the growth of beneficial bacterial populations [33]. On the other hand, MAC showed no effect on OM digestibility. As the variables are statistically independent, this occurred due to high variability and the smaller number of studies to carry out the meta-analysis for OM than DM digestibility. Despite the high heterogeneity, the meta-regression was not able to adequately explain the source of variation, except for NDF digestibility. The analysis indicated that the intake of starch and also of protein (g/kg of BW) reduces the MAC ES on NDF digestibility. It is possible that starch intake reduces the MAC effect on NDF digestibility because the rapid fermentation of starch decreases rumen pH, creating a less favorable environment for cellulolytic bacteria that are responsible for fiber digestion [35].

### 4.3. Blood Parameters

Despite being considered small (ES = 0.170), a significant effect on serum glucose level was detected due to MAC addition. Changes in this variable are related to the increase in propionate in the rumen and absorption by the epithelium, resulting in greater hepatic glucose synthesis [36]. Although our study observed greater protein digestibility for the treated group, plasma urea was not influenced by MAC addition. The concentration of urea N in plasma is used as an indicator to evaluate the protein status or protein nutrition of ruminants [37]. Despite the high heterogeneity for plasma urea, none of the covariates tested were adequate to explain the between-studies variance. Animals fed MAC also had lower levels of NEFA (ES = −0.404), which indicates less mobilization of body fat. This is an important answer because the level of NEFA in plasma correlates with the negative energy balance in early lactation cows, which allows this variable to be used as an indicator of energy balance [38]. It is important to point out that the small number of studies on most blood parameters may result in less precise estimates of the overall or subgroup effects and heterogeneity associated with these variables [22]. In addition, the results for lactate and NEFA are based on subgroups with fewer than three comparisons and should therefore be interpreted with caution due to limited reliability.

## 5. Conclusions

Malic acid addition appears to modulate ruminal fermentation and nutrient utilization in cattle, with responses influenced by its chemical form. The observed improvements in rumen environment, fiber and protein digestibility, and certain blood metabolites indicate the potential of malic acid, particularly its salt form, to enhance digestive efficiency and metabolic balance. However, the variability in responses across studies may indicate the importance of diet composition as a moderating factor, which should be considered when including this feed additive in cattle diets.

## Figures and Tables

**Figure 1 animals-15-02177-f001:**
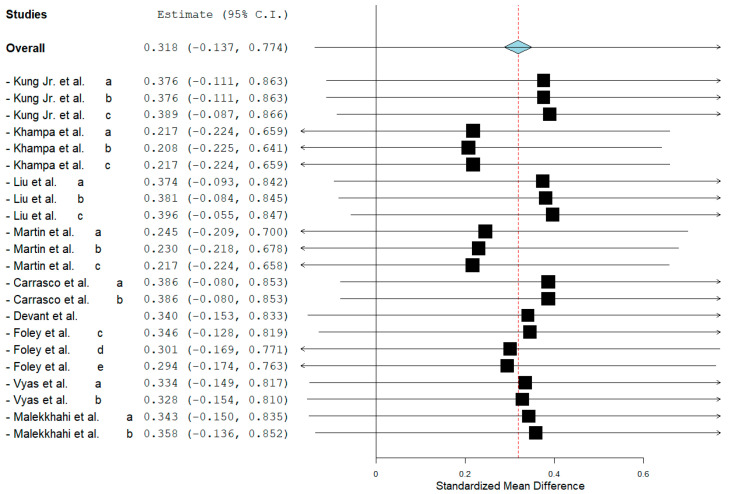
Leave-one-out sensitivity analysis for the effect of the addition of malic acid or malate on the rumen pH of cattle. Each line represents the overall effect estimate recalculated after excluding one study at a time. The diamond shows the new standardized mean difference (SMD) and its 95% confidence interval. The limited variation in estimates indicates that no individual study had a disproportionate influence on the overall result [6,7,8,9,12,15,16,19,20]. Lowercase letters indicate different experiments/comparisons within a study.

**Figure 2 animals-15-02177-f002:**
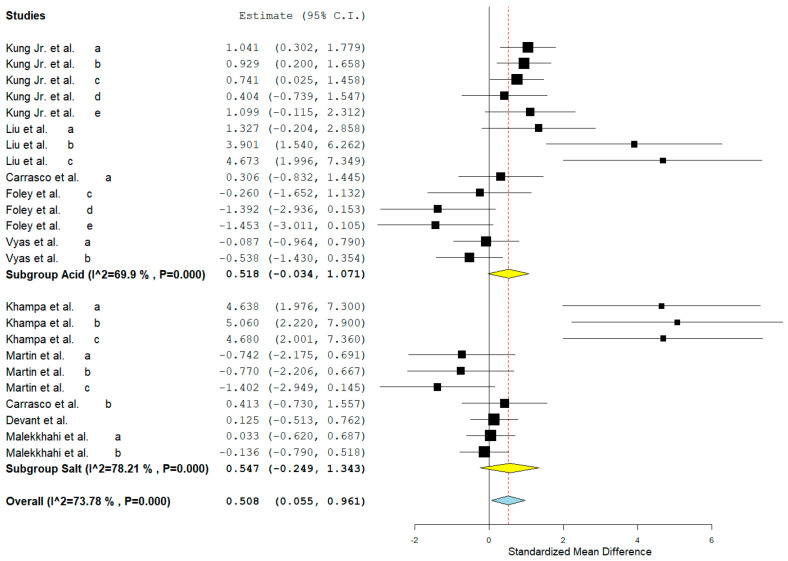
Forest plot of the effect of malic acid or malate addition on total volatile fatty acids in cattle. When the diamond was presented to the left of the central line (standardized mean) without touching it, the effect was considered to be negative, favoring control. When presented to the right of the center line, the effect was considered positive, in favor of the feed additive [6,7,8,9,12,15,16,19,20]. Lowercase letters indicate different experiments/comparisons within a study.

**Figure 3 animals-15-02177-f003:**
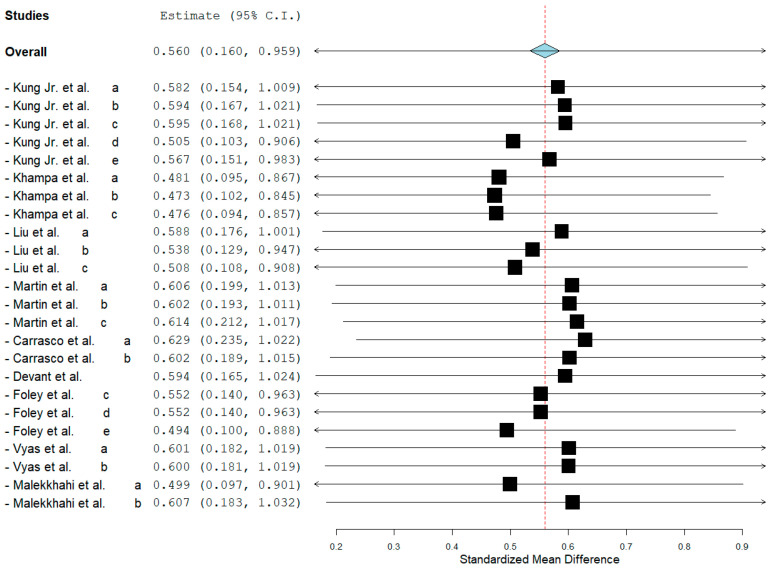
Leave-one-out sensitivity analysis for the effect of malic acid or malate addition on rumen propionate of cattle. Each line represents the overall effect estimate recalculated after excluding one study at a time. The diamond shows the new standardized mean difference (SMD) and its 95% confidence interval. The limited variation in estimates indicates that no individual study had a disproportionate influence on the overall result [6,7,8,9,12,15,16,19,20]. Lowercase letters indicate different experiments/comparisons within a study.

**Figure 4 animals-15-02177-f004:**
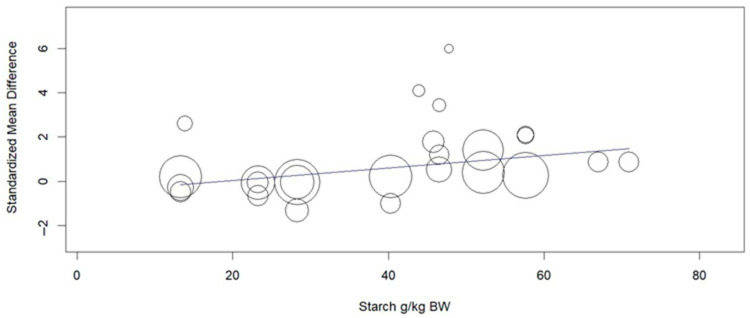
Meta-regression of the effect of starch intake (g/kg BW) on the standardized mean difference of malate or malic acid addition on propionate in the rumen of cattle.

**Table 1 animals-15-02177-t001:** Chemical form, cereal and main forage, dose, and calculated composition of the total ration mixed in experiments with cattle feeding malic acid or malate.

Author	Form	Main Cereal	Main forage	Dose (g/day)	CP (%)	NDF (%)	ADF (%)	Starch (%)	EE (%)
Kung Jr. et al. [6] a	Acid	Corn	Corn silage	70.00	11.18	24.71	13.95	35.77	1.99
Kung Jr. et al. [6] b	Acid	Corn	Corn silage	105.00	11.18	24.71	13.95	35.77	1.99
Kung Jr. et al. [6] c	Acid	Corn	Corn silage	140.00	11.18	24.71	13.95	35.77	1.99
Kung Jr. et al. [6] d	Acid	Corn	Corn silage	42.00	8.76	25.54	13.76	46.78	2.77
Kung Jr. et al. [6] e	Acid	Corn	Corn silage	84.00	8.76	25.54	13.76	46.78	2.77
Khampa et al. [7] a	Salt	Cassava	Rice straw	9.00	8.61	41.14	23.86	34.90	3.51
Khampa et al. [7] b	Salt	Cassava	Rice straw	18.00	8.61	41.14	23.86	34.90	3.51
Khampa et al. [7] c	Salt	Cassava	Rice straw	27.00	8.61	41.14	23.86	34.90	3.51
Liu et al. [8] a	Acid	Corn	Corn straw	70.20	8.29	55.82	21.85	14.78	1.72
Liu et al. [8] b	Acid	Corn	Corn straw	140.40	8.29	55.82	21.85	14.78	1.72
Liu et al. [8] c	Acid	Corn	Corn straw	210.60	8.29	55.82	21.85	14.78	1.72
Martin et al. [9] a	Salt	Corn	Cottonseed hulls	27.00	11.39	19.10	9.93	49.60	2.95
Martin et al. [9] b	Salt	Corn	Cottonseed hulls	54.00	11.39	19.10	9.93	49.60	2.95
Martin et al. [9] c	Salt	Corn	Cottonseed hulls	80.00	11.39	19.10	9.93	49.60	2.95
El-Zaiat et al. [10]	Acid	Corn	Corn silage	30.00	17.16	32.29	19.06	36.70	5.60
Carrasco et al. [12] a	Acid	Barley	Barley straw	9.38	16.61	21.59	8.35	37.04	9.76
Carrasco et al. [12] b	Salt	Barley	Barley straw	9.12	16.61	21.59	8.35	37.04	9.76
Sniffen et al. [14]	Salt	Corn	Corn silage	50.00	18.20	31.80	21.40	29.40	2.70
Devant et al. [15]	Salt	-	-	84.00	14.31	32.84	16.93	30.16	3.16
Foley et al. [16] a	Acid	Barley	Silage	34.00	15.60	23.10	13.80	28.10	2.50
Foley et al. [16] b	Acid	Barley	Silage	65.40	15.60	23.10	13.80	28.10	2.50
Foley et al. [16] c	Acid	Barley	Silage	32.38	15.57	23.09	13.82	28.12	2.47
Foley et al. [16] d	Acid	Barley	Silage	64.85	15.57	23.09	13.82	28.12	2.47
Foley et al. [16] e	Acid	Barley	Silage	98.25	15.57	23.09	13.82	28.12	2.47
Wang et al. [17] a	Acid	Corn	Corn silage	70.00	16.50	42.40	27.10	31.70	1.50
Wang et al. [17] b	Acid	Corn	Corn silage	140.00	16.50	42.40	27.10	31.70	1.50
Wang et al. [17] c	Acid	Corn	Corn silage	210.00	16.50	42.40	27.10	31.70	1.50
Hernández et al. [18] a	Salt	Barley	Barley straw	30.80	13.83	37.30	16.62	28.51	3.87
Hernández et al. [18] b	Acid	Barley	Barley straw	26.80	13.83	37.30	16.62	28.51	3.87
Hernández et al. [18] c	Salt	Barley	Barley straw	28.40	13.83	37.30	16.62	28.51	3.87
Vyas et al. [19] a	Acid	Barley	Barley silage	89.00	9.74	16.86	6.57	45.32	1.57
Vyas et al. [19] b	Acid	Barley	Barley silage	177.00	9.74	16.86	6.57	45.32	1.57
Malekkhahi et al. [20] a	Salt	Corn	Corn silage	80.00	17.69	27.64	16.66	29.90	2.23
Malekkhahi et al. [20] b	Salt	Corn	Corn silage	80.00	20.93	32.50	18.25	45.53	2.74

CP, crude protein; NDF, neutral detergent fiber; ADF, acid detergent fiber; EE, ether extract; lowercase letters indicate different experiments/comparisons within a study.

**Table 2 animals-15-02177-t002:** Summary of meta-analysis (effect size) of malate or malic acid addition on ruminal parameters of cattle.

Variable	NP	Form	NC	ES (CI)	ES *p*-Value	I^2^	Het *p*-Value
pH		Salt	11	1.420 (0.558; 2.282)	0.00	80.40	<0.01
	9	Acid	12	−0.310 (−0.698; 0.079)	0.12	38.18	0.09
		Overall	23	0.310 (−0.137; 0.774)	0.17	72.95	<0.01
Acetate		Salt	11	−0.592 (−1.381; 0.196)	0.14	72.26	<0.01
	9	Acid	14	0.167 (−0.386; 0.720)	0.55	78.08	<0.01
		Overall	25	−0.120 (−0.584; 0.345)	0.61	76.34	<0.01
Butyrate		Salt	11	−0.356 (−1.040; 0.328)	0.31	72.84	<0.01
	9	Acid	14	−0.058 (−0.717; 0.601)	0.86	78.39	<0.01
		Overall	25	−0.178 (−0.653; 0.297)	0.46	76.69	<0.01
Propionate		Salt	11	0.756 (−0.075; 1.588)	0.08	80.10	<0.01
	9	Acid	14	0.472 (0.066; 0.879)	0.02	48.17	0.02
		Overall	25	0.560 (0.160; 0.959)	0.01	67.31	<0.01
Lactate		Salt	6	0.337 (−0.517; 1.191)	0.44	67.60	0.01
	5	Acid	6	−0.621 (−1.512; 0.270)	0.17	74.94	<0.01
		Overall	12	−0.113 (−0.711; 0.485)	0.71	70.76	<0.01
ACT:PRP		Salt	9	−1.327 (−2.683; 0.030)	0.06	82.04	<0.01
	6	Acid	6	−1.109 (−2.470; 0.252)	0.11	83.70	<0.01
		Overall	15	−1.130 (−2.028; −0.232)	0.01	81.68	<0.01
NH_3_N		Salt	8	0.161 (−0.170; 0.492)	0.34	0.99	0.42
	7	Acid	12	−0.089 (−0.560; 0.381)	0.71	47.12	0.04
		Overall	20	0.079 (−0.227; 0.385)	0.61	36.62	0.05
Total VFA		Salt	11	0.547 (−0.249; 1.343)	0.18	78.21	<0.01
	9	Acid	14	0.518 (−0.034; 1.071)	0.07	69.90	<0.01
		Overall	25	0.508 (0.055; 0.961)	0.03	73.78	<0.01

NP, number of papers; NC, number of comparisons; ES, effect size; CI, confidence interval; VFA, volatile fatty acids; NH_3_N, ammonia nitrogen; ACT:PRP, acetate: propionate ratio.

**Table 3 animals-15-02177-t003:** Summary of meta-analysis (effect size) of malate or malic acid addition on blood parameters and diet digestibility of cattle.

Trait	NP	Form	NC	ES (CI)	*p*-Value	I^2^	Het *p*-Value
Blood parameters							
Glucose		Salt	7	0.163 (−0.132; 0.457)	0.28	0.00	0.88
	8	Acid	9	0.173 (−0.034; 0.379)	0.10	0.49	0.43
		Overall	16	0.170 (0.002; 0.338)	0.05	0.00	0.78
Urea		Salt	7	0.028 (−0.385; 0.441)	0.89	45.12	0.11
	6	Acid	8	−0.109 (−0.413; 0.194)	0.48	53.35	0.04
		Overall	15	−0.033 (−0.279; 0.212)	0.79	47.24	0.03
Lactate		Salt	7	−0.060 (−0.956; 0.836)	0.90	82.63	<0.01
	4	Acid	2	−1.661 (−2.690; −0.361)	0.01	57.23	0.13
		Overall	9	−0.490 (−1.316; 0.337)	0.25	83.31	<0.01
NEFA		Salt	2	−0.024 (−0.597; 0.550)	0.94	0.00	0.94
	3	Acid	7	−0.626 (−1.065; −0.187)	0.01	0.00	0.47
		Overall	9	−0.404 (−0.759; −0.049)	0.03	3.56	0.40
β-hidroxibutirate		Salt	2	0.532 (−0.769; 1.832)	0.42	78.81	0.03
	3	Acid	7	−0.260 (−1.172; 0.652)	0.58	75.68	0.01
		Overall	9	−0.018 (−0.742; 0.706)	0.96	75.20	<0.01
Digestibility							
Dry matter		Salt	5	−0.084 (−0.575; 0.407)	0.74	0.00	0.95
	6	Acid	8	0.940 (0.229; 1.651)	0.01	73.01	0.01
		Overall	13	0.547 (0.027; 1.067)	0.04	78.74	<0.01
Organic matter		Salt	4	0.056 (−0.435; 0.546)	0.82	0.00	0.99
	6	Acid	5	0.694 (−0.217; 1.604)	0.14	53.15	0.07
		Overall	9	0.308 (−0.148; 0.764)	0.19	21.24	0.25
Protein		Salt	5	1.168 (0.217; 2.118)	0.02	52.70	0.10
	6	Acid	8	0.215 (−0.197; 0.627)	0.31	0.00	0.97
		Overall	13	0.422 (0.099; 0.745)	0.01	0.00	0.47
NDF		Salt	5	1.537 (0.277; 2.797)	0.02	77.39	0.00
	6	Acid	6	−0.085 (−0.576; 0.406)	0.73	0.00	0.94
		Overall	11	0.699 (−0.007; 1.406)	0.05	67.29	0.01
ADF		Salt	4	0.547 (0.042; 1.051)	0.03	0.00	0.45
	6	Acid	8	0.654 (−0.078; 1.387)	0.08	60.62	0.01
		Overall	12	0.635 (0.148; 1.121)	0.01	46.49	0.03

NP, number of papers; NC, number of comparisons; ES, effect size; CI, confidence interval; NEFA, non-esterified fatty acids; NDF, neutral detergent fiber; ADF, acid detergent fiber.

**Table 4 animals-15-02177-t004:** Meta-regression of the effect of malic acid or malate addition on ruminal and blood parameters and digestibility of dietary fractions determined with cattle.

Variables			Covariates, g/kg BW			
	NP	NC	NDF	ADF	Starch	Organic Acid
Rumen parameters						
pH	9	23	2.063 − 0.051x *	0.908 − 0.142	0.584 − 0.004x	0.343 + 0.548x
Acetate	9	25	0.126 − 0.008x	0.464 −0.173x	−1.726 + 0.039x **	−0.455 + 2.690x
Butyrate	9	25	−1.475 + 0.038x	−1.155 + 0.252x	−0.649 + 0.009x	−1.121 + 7.116x
Propionate	9	25	0.239 + 0.009x	0.357 + 0.055x	−0.518 + 0.028x **	0.871 − 2.894x
Lactate	5	12	1.250 − 0.041x *	0.684 − 0.242x	−1.462 + 0.031x ^T^	−0.547 + 3.758x
Acetate:propionate	6	15	1.887 − 0.118x *	3.057 − 1.463x **	−7.483 + 0.156x **	−3.000 + 10.725x
NH_3_N	7	20	1.337 − 0.033x **	0.854 − 0.176x	−0.032 + 0.002x	0.262 − 1.384x
Total VFA	9	25	−0.034 + 0.019x	0.632 − 0.013x	−1.033 + 0.042x *	1.258 − 5.318x
Blood parameters						
Urea	6	15	−0.074 + 0.001x	−0.053 + 0.005x	0.370 − 0.007x	0.007 − 0.190x
Digestibility						
Dry matter	6	13	−0.374 + 0.023x ^T^	0.418 + 0.021x	0.673 − 0.004x	1.067 − 6.511x
Protein	6	13	0.613 − 0.004x	0.336 + 0.016x	0.483 − 0.002x	0.560 − 1.665x
NDF	6	11	0.135 + 0.013x	1.309 − 0.171x ^T^	1.375 − 0.025x *	1.530 − 13.733x ^T^
ADF	6	12	0.550 − 0.002x	0.695 − 0.039x	1.167 − 0.016x	0.795 − 3.606x

NP, number of papers; NC, number of comparisons; ES, effect size; CI, confidence interval; NEFA, non-esterified fatty acids; NDF, neutral detergent fiber; ADF, acid detergent fiber. * *p* < 0.05; ** *p* < 0.01; ^T^ tendency; VFA, volatile fatty acids.

## Data Availability

No new data were created or analyzed in this study. All data supporting the findings are publicly available in the original publications referenced in the manuscript.

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
