# Peer review of "Exploratory Meta-Analysis of the Effect of Malic Acid or Malate Addition on Ruminal Parameters, Nutrient Digestibility, and Blood Characteristics of Cattle"

_animals, 2025, doi:10.3390/ani15152177_

Round 1

Reviewer 1 Report

Comments and Suggestions for Authors

Comments and Suggestions for Authors

After reviewing the manuscript entitled “Meta-analysis of the effect of malic acid or malate supplementation on ruminal parameters, nutrient digestibility and blood characteristics of cattle”, the following suggestions were made it. The manuscript contains several design errors, which render it of low reliability and scientific quality. Therefore, the current manuscript should not be published in a prestigious journal such as Animals and should be rejected without further consideration. The individual corrections are shown below:

Title

Line 4: Beef cattle or dairy cattle? Please be specific.

Simple Summary

Line 16: Beef cattle or dairy cattle? Please be specific

Abstract

Line 23: Beef cattle or dairy cattle? Please be specific

Line 23: The effect size (ES) is not a method; it is the unit of measurement used to measure the treatment effect in meta-analytic studies. Therefore, authors should specify the specific statistical method (there are several common statistical methods in meta-analytic studies) used to analyze the data. For example, the DerSimonian-Laird fixed-effects or random-effects model and the restricted maximum likelihood model, among others. This information is essential to verify that the authors correctly selected the analysis model.

Line 24: Subgroup analyses and meta-regression are not sensitivity analyses. Therefore, authors should conduct the aforementioned sensitivity analyses correctly. Subgroup analyses and meta-regression are complementary methods that allow for the identification of sources of heterogeneity between studies. Therefore, the usefulness of these two methods differs.

Lines 25-27: The description of the subgroups used is not relevant in the abstract. Please delete these lines and instead briefly describe the methodology used in the search and selection of the articles included in the current meta-analysis. Also, mention the range of years and the electronic databases used in the information searches. Finally, indicate the number of scientific articles included in the meta-analysis database. These features will help the reader understand whether the authors used valid procedures and a database with the required number of articles to be considered scientifically valid.

Line 28: The abbreviation MAC must be used the first time malic acid is written. In this case, it was first written on line 22 of the abstract.

Lines 34-35: Please specify which treatment (malic acid or malate) increased serum glucose levels and decreased serum NEFA levels.

Lines 36-39: The conclusions should be rewritten because, in their current form, they read more like a description of the results, which were described in the previous lines of the abstract. The conclusions should be rewritten in the present tense, showing an interpretation of the results and their implications rather than a description of the changes observed with the evaluated treatments.

Introduction

Lines 47-48: Before mentioning malic acid as an alternative to the use of ionophores as a livestock feed additive, the authors should provide information on other additives that have been commonly used to replace ionophores. From an economic and biological perspective, the authors should then justify why malic acid should be studied instead of any other currently available feed additives.

Line 51: Before beginning to describe the effects of malic acid reported by other studies, the authors should indicate the biological properties that give it potential as a ruminant feed additive.

Lines 51-54: The background information reported by the authors is limited and does not justify conducting a meta-analysis. A meta-analysis is often conducted when there is extensive conflicting information on a topic. However, these lines provided by the authors do not reveal conflicting information. Therefore, these lines should be supplemented. First, the authors should include information on studies that have observed positive, negative, and neutral effects of malic acid use in cattle, indicating the direction of the effect (positive, negative, or neutral) in each case. Likewise, for each study added and cited, the authors should indicate the doses of malic acid or malate used, the periods of supplementation, the species of cattle, the physiological stage, and the type of diet (high-forage or high-concentrate) used in each study. This information will help readers understand the current inconsistencies regarding the effects of malic acid as a livestock feed additive, identify factors influencing the variability of results, and justify conducting the current meta-analysis. The information mentioned above should be focused on the animal species evaluated in the current study, which should be clarified (dairy cows or beef cattle?).

Lines 57-60: To complement this information, authors should add a few lines justifying the use of meta-analysis instead of a traditional narrative review. For example, the advantages and disadvantages of both methods (meta-analysis and traditional narrative review) should be highlighted.

Line 61: After adding the information requested in the previous comments, the authors should add a clear hypothesis of the meta-analysis before the objective.

Material and methods

Lines 68-69: According to Borenstein (2019), abstracts, master's dissertations, and doctoral theses are strictly prohibited in meta-analytic studies because they represent gray literature (literature that has not been peer-reviewed and is, in most cases, duplicated because it is published in scientific articles). Therefore, the authors' search methodology is incorrect and reflects a lack of knowledge about meta-analytic methodology. It is strictly necessary to include only articles published in peer-reviewed scientific journals.

Borenstein, M., 2019. Common mistakes in meta-analysis and how to avoid them. Biostat, Inc., Englewood, NJ, USA.

Lines 66-73: To ensure the meta-analysis's replicability, authors should include the year range and language used in the information search. They should also include a PRISMA flowchart showing the flow of information from identification to inclusion in the meta-analysis database.

Lines 74-77: Add the equations and scientific references used to justify the estimation of the parameters mentioned. Were the data reported in graphs not included in the meta-analysis database? The current results are invalid if they were not included, and all missing data should be included.

Line 78: Clearly indicate the characteristics that the control and MAC treatments should have..

Line 80: Before mentioning the information extracted from the selected articles, the authors should list the exclusion and inclusion criteria used in the current meta-analysis.

Line 86: Table 1 only contains 13 studies not 19. Please review and correct.

Line 86: The authors should list the response variables evaluated in the meta-analysis in a separate paragraph.

Lines 92-97: The effect size (ES) is not a method; it is the unit of measurement used to measure the treatment effect in meta-analytic studies. Before beginning the description of the procedures used, authors should specify the statistical software used (and the packages within the software, if applicable), including the version, and should cite at least one scientific reference justifying the validity of the software used for meta-analytic studies. Likewise, they should clarify what the effect size refers to and explain in greater detail how it was calculated. They should justify why the effect size was chosen instead of other units of measurement available for meta-analytic studies (this could be justified depending on the data type).

Lines 98-100: Subgroup analyses, and particularly meta-regression, are used to assess heterogeneity between studies, not to discuss results. Please correct. Likewise, subgroup analyses should only be performed in strata with at least three comparisons to ensure scientifically valid results.

Line 103: This methodology is incorrect and can lead to the detection of false positives in the results. Meta-regression should only be used for response variables reported in at least 10 different articles, not in 10 different comparisons because those 10 comparisons could all be found in a single article. This would make it impossible to assess heterogeneity between studies.

Line 107: In meta-analytic studies, it is absolutely mandatory to assess the presence of publication bias using at least one statistical test (e.g., Begg's adjusted rank correlation, Egger's regression asymmetry, Rosenberg's safety number, among others). In case of publication bias, authors should apply the trim-and-fill method of Duval and Tweedie (2000) to estimate the possible number of missing observations. Without publication bias analyses, the current meta-analysis should be immediately rejected.

Results

Lines 108-185: The interpretation of the results is correct. However, the entire section is poorly organized. First, the authors should report and interpret the overall malic acid/malate supplementation results. After showing the main effects results for each group of response variables, the results of subgroup analyses should be presented. In their current form, the subgroup analysis figures report duplicate information shown in the tables, where grouping was done by malic acid or malate. Therefore, the figures should be removed to avoid duplication of information. Likewise, Table 3 contains results from subgroups with fewer than three comparisons, which should be removed because they are scientifically unreliable. Furthermore, the results of the publication bias tests suggested in a previous commentary should be added to this results section. Finally, all abbreviations shown within Tables 2 and 3 should be defined in the table footnotes. Furthermore, authors must specify the species in which the malic acid/malate treatments were evaluated. Were dairy cows or beef cattle used? This information must be specified throughout the manuscript, including in the tables of the results.

Discussion

Lines 187-197: These lines repeat the hypotheses, objectives, and results of the current study, which is incorrect. Therefore, this paragraph should be removed from this section. The discussion section should focus on comparing the results obtained with those previously reported by other authors. Likewise, the discussion section should focus exclusively on explaining the results obtained based on the biochemical, physiological, microbiological, or nutrigenomic mechanisms of action of malic acid/malate.

Lines 199-302: The discussion provided by the authors is acceptable for the manuscript. However, all ES values ​​added in parentheses should be removed from this section because they were previously reported in the results section. For example, the following should be removed from line 296: “(ES= -0.404)”. In addition, after correcting the results section, the authors should make corresponding changes to this discussion section. Throughout the discussion, the authors should have added more focused information on the animal species evaluated, which was not specified (beef cattle or dairy cows?).

Conclusions

Lines 304-312: The conclusions section should be removed and rewritten because, in its current form, it only briefly repeats the description of the results, which is incorrect. The new conclusions should be rewritten in the present tense and should show a comprehensive interpretation and implication of the groups of response variables evaluated, rather than repeating the observed effect on each response variable.

Author Response

We would like to thank the reviewers for their constructive comments, which have contributed to improving the quality of our manuscript. Whenever possible, we have addressed all the suggestions and requests made. The specific editorial recommendations were incorporated directly into the revised manuscript. In this document, we aim to respond to the main questions and concerns raised by the reviewers.

COMMENT Line 16: Beef cattle or dairy cattle? Please be specific

R: We chose to use the general term cattle in the title because studies involving both beef and dairy cattle were included in the analysis. Additionally, as productive performance parameters were not evaluated in this study, we did not consider it necessary to separate the categories in the title.

Comments for abstract

The suggested changes to the abstract were addressed as much as possible

Comments for introduction

Regarding the comments on the introduction, we have partially addressed the reviewer’s suggestions. We believe that many of the requested elements were already present in the original version, and we made additional adjustments to clarify and emphasize those points more succinctly, while maintaining the focus and conciseness of the section.

COMMENT Lines 68-69: According to Borenstein (2019), abstracts, master's dissertations, and doctoral theses are strictly prohibited in meta-analytic studies because they represent gray literature (literature that has not been peer-reviewed and is, in most cases, duplicated because it is published in scientific articles). Therefore, the authors' search methodology is incorrect and reflects a lack of knowledge about meta-analytic methodology. It is strictly necessary to include only articles published in peer-reviewed scientific journals.

R: We understand the concern regarding the inclusion of unpublished data, such as theses and dissertations, in meta-analyses. However, we do not consider this a prohibitive practice, especially when appropriate quality criteria are applied. Several meta-analyses in the animal science field, including those conducted by well-recognized authors such as D. Sauvant (Eugène et al., 2004) and J. Lean (Rodney et al., 2015), have included grey literature in their databases.

COMMENT: Lines 74-77: Add the equations and scientific references used to justify the estimation of the parameters mentioned. Were the data reported in graphs not included in the meta-analysis database? The current results are invalid if they were not included, and all missing data should be included.

R: No data were excluded from the database. All information presented was extracted using, and data transformations (such as conversions from standard error, confidence intervals, and p-values to standard deviations) were performed following the recommendations of the Cochrane Handbook for Systematic Reviews of Interventions (Chapter 6.5).

COMMENT Subgroup analyses, and particularly meta-regression, are used to assess heterogeneity between studies, not to discuss results. Please correct. Likewise, subgroup analyses should only be performed in strata with at least three comparisons to ensure scientifically valid results.

R: We decided to retain all subgroup information in the article, even those with only two comparisons, as their presentation is not uncommon in the literature. However, we added a note to the text warning about the low reliability of results obtained from subgroups with fewer than three comparisons.

COMMENT In meta-analytic studies, it is absolutely mandatory to assess the presence of publication bias using at least one statistical test (e.g., Begg's adjusted rank correlation, Egger's regression asymmetry, Rosenberg's safety number, among others). In case of publication bias, authors should apply the trim-and-fill method of Duval and Tweedie (2000) to estimate the possible number of missing observations. Without publication bias analyses, the current meta-analysis should be immediately rejected.

R: The leave-one-out analysis was conducted to assess the influence of individual studies on the overall effect estimate in the meta-analysis. The results of this analysis have been incorporated into the manuscript.

COMMENT: This methodology is incorrect and can lead to the detection of false positives in the results. Meta-regression should only be used for response variables reported in at least 10 different articles, not in 10 different comparisons because those 10 comparisons could all be found in a single article. This would make it impossible to assess heterogeneity between studies.

R: We understand the concern and agree that, ideally, meta-regression analyses should be based on data from at least 10 different studies to properly assess between-study heterogeneity. Whenever possible, we strive to meet this criterion. However, while strongly recommended, having 10 separate articles is not a strict requirement. As noted by Austin and Steyerberg (2015), in the context of linear regression, a minimum of only two subjects per variable (SPV) may be sufficient for an adequate estimation of regression coefficients. Furthermore, similar approaches to ours—where the number of comparisons rather than the number of articles served as the basis for meta-regression—can be found in studies related to animal science, such as those by Toledo et al. (2019) and Lean et al. (2014), supporting the methodological framework adopted in our analysis.

Austin, P. C., & Steyerberg, E. W. (2015). The number of subjects per variable required in linear regression analyses. Journal of Clinical Epidemiology, 68(6), 627–636.

Toledo, T. D. S. D. et al. (2019). The effect of litter materials on broiler performance: a systematic review and meta-analysis. British Poultry Science, 60(6), 605–616.

Lean, I. J., Thompson, J. M., & Dunshea, F. R. (2014). A meta-analysis of zilpaterol and ractopamine effects on feedlot performance, carcass traits and shear strength of meat in cattle. PLOS ONE, 9(12), e115904.

Reviewer 2 Report

Comments and Suggestions for Authors

The aim of the study was to determine, through meta-analysis, the effect of malic acid or malate supplementation on some rumen and blood parameters of cattle. The work corresponds to the scope of the journal.
The manuscript is laid out clearly, corresponds to the research topics, and is well structured.
The cited references are mostly publications older than 5 years. It is necessary to strengthen the manuscript with more recent references from the last 5 years. The relevance of the meta-analysis is not fully reflected and justified in the introduction.
The manuscript is scientifically sound and the experimental design is appropriate to test the hypothesis.
The results of the manuscript are generally reproducible based on the information given in the methods section. But a description of the calculation of exactly how the authors determined the standard deviation when it was not available in the primary materials should be added. Provide formulas. Also the formula for calculating the effect size (effect size method).
What exactly were the commercial products used? Were they the same additives? The dosages? Their characterization should be stated. The characteristics of the cattle taken for the studies (breed, age, stage of lactation, productivity level) are not fully understood. A very wide time range of studies is taken. Genetic characteristics should be taken into account.
Figures/tables/images/charts are a reflection of the analysis of the research material presented. Data are presented correctly. Figures allow for quick interpretation and understanding of data. Data are interpreted appropriately and consistently throughout the manuscript. Statistical analysis of the data has been performed for all material presented.
The conclusions are generally consistent with the evidence and arguments presented but are consistent with existing knowledge, new mechanisms and knowledge gained from the meta-analysis should be emphasized. Recommendations for the use of the supplements studied are needed.

Author Response

We would like to thank the reviewer for their constructive comments, which have contributed to improving the quality of our manuscript. Whenever possible, we have addressed all the suggestions and requests made. The specific editorial recommendations were incorporated directly into the revised manuscript.

However, not all suggested changes were applied, as in some cases the recommendations from different reviewers pointed in conflicting directions. Nevertheless, if there are any points that still require clarification, we would be happy to provide further explanation.

Reviewer 3 Report

Comments and Suggestions for Authors

I would like to begin by congratulating the authors for the excellent work they produced in this meta-analysis. Malates are important feed additives for diet balance, and there is a lot of information in isolated papers; however, when they are combined to build a final idea, we have much more science. Before recommending publication, I will make notes and add to the text regarding structure, clarity, and statistical interpretations to strengthen the manuscript.

Title

Terminology: I would not use the word supplementation in the title and text when referring to malate. Supplementation increases the supply of something that is already in the basal diet, which is not the case with malate, which has a different source. I suggest that the authors use the term “dietary malatae” instead of “malate supplementation.” Rectify throughout the manuscript.

Abstract

The authors should add a short introductory sentence to the abstract before stating the aims of the study.

Introduction

The introduction should be improved.

I invite the authors to add a hypothesis for their study before the aims of the study.

Methods

Normally, master's dissertations and PhD theses should not be included in the meta-analysis, as there could be repetition of data in PhD theses with papers, and the dissertations are not peer reviewed.

For this “These were not provided directly, they were calculated using the measures presented in the paper, such as standard error of the mean, coefficients of variation and others,” provide the equation you used to calculate the standard deviation. I invite you to cite this from the following meta-analysis:

Meta-Analysis of Dietary Tannins in Small Ruminant Diets: Effects on Growth Performance, Serum Metabolites, Antioxidant Status, Ruminal Fermentation, Meat Quality, and Fatty Acid Profile. Animals, 15(4), 596. https://doi.org/10.3390/ani15040596

For this paragraph “Independent experiments in the same study were included as a new comparison. Similarly, malate doses in the same trial were included as a new comparison.” There was a repetition as malate doses were considered as independent experiments.

Line 86:19 studies are a small number for meta-analysis. Moreover, the distribution between species (e.g., dairy vs. beef), production stage (e.g., lactating vs. growing), and diet types (e.g., forage: concentrate ratios) can be more clearly summarized.

There was no mention of a funnel plot or Egger’s test to evaluate potential publication bias. I invite the authors to include a brief comment or figure to demonstrate that publication bias was considered (even if tests were limited by the number of studies per outcome). I invite them to cite the following paper:

Microalgae supplementation improves goat milk composition and fatty acid profile: a meta-analysis and meta-regression. Archives Animal Breeding, 68(1), 223-238. https://doi.org/10.5194/aab-68-223-2025

Table 1: Convert A, B, C, … in the first column to a, b, c,…

Correct the error in “Saltt”

What do you mean by the main cereal ? main forage ? Are these sources of malate? If malate was present in only one of them, only the source of malate should be reported.

Results

Some outcomes (e.g., acetate: propionate ratio, total VFA, lactate) showed I² > 80%, yet definitive conclusions were drawn. I invite authors to temper conclusions in cases where heterogeneity is high and explain how this affects confidence in the pooled results.

Forest plots are informative, but can be accompanied by clearer figure legends (e.g., sample sizes, confidence intervals). Ensure that all figures include clear axis labels and define symbols (diamond shapes, subgroup lines) consistently.

Discussion

Line 286: I invite authors to cite the following paper, which details the difference between two different starch sources on their digestibility.

Growth performance, carcass characteristics, fatty acid profile, and meat quality of male goat kids supplemented by alternative feed resources: bitter vetch and sorghum grains. Archives Animal Breeding, 67(4), 481-492. https://doi.org/10.5194/aab-67-481-2024

Lines 287–302: Since the number of studies on blood metabolites is low, caution should be used in drawing strong conclusions here.

Comments on the Quality of English Language

The English could be improved to express the research more clearly. Moderate revision is needed.

Line 57: correct to “to allow the results…”

Line 83: “diet chemical composition” should be “the diet’s chemical composition”.

Line 205: “the ES of 1,420” → revise to “the effect size (ES) of 1.420”.

Line 236: ‘data not shown’ is vague; either cite supplementary material or omit.

Comments on the Quality of English Language

The Quality of English should be improved. 

Author Response

We would like to thank the reviewer for their constructive comments, which have contributed to improving the quality of our manuscript. Whenever possible, we have addressed all the suggestions and requests made. The specific editorial recommendations were incorporated directly into the revised manuscript.

However, not all suggested changes were applied, as in some cases the recommendations from different reviewers pointed in conflicting directions. Nevertheless, if there are any points that still require clarification, we would be happy to provide further explanation.

COMMENT: Normally, master's dissertations and PhD theses should not be included in the meta-analysis, as there could be repetition of data in PhD theses with papers, and the dissertations are not peer reviewed.

R: We understand the concern regarding the inclusion of unpublished data, such as theses and dissertations, in meta-analyses. However, we do not consider this a prohibitive practice, especially when appropriate quality criteria are applied. Several meta-analyses in the animal science field, including those conducted by well-recognized authors such as D. Sauvant (Eugène et al., 2004) and J. Lean (Rodney et al., 2015), have included grey literature in their databases.

COMMENT: There was no mention of a funnel plot or Egger’s test to evaluate potential publication bias. I invite the authors to include a brief comment or figure to demonstrate that publication bias was considered (even if tests were limited by the number of studies per outcome). I invite them to cite the following paper:
R: The leave-one-out analysis was conducted to assess the influence of individual studies on the overall effect estimate in the meta-analysis. The results of this analysis have been incorporated into the manuscript.

Round 2

Reviewer 3 Report

Comments and Suggestions for Authors

I invite authors to respond point-by-point to all the comments I adressed.

Comments on the Quality of English Language

The quality of English should be improved. 

Author Response

We apologize for the oversight. The responses were submitted to the editor and forwarded to the reviewer.

Round 3

Reviewer 3 Report

Comments and Suggestions for Authors

I appreciate the authors’ detailed point-by-point response to my comments. After reviewing their replies and the revised manuscript, I am satisfied that all my concerns have been adequately addressed.

I consider the manuscript suitable for publication in its current form.

Author Response

We thank the reviewer for the comment